# Global patterns of tree density are contingent upon local determinants in the world's natural forests

Jaime Madrigal-González [1,2 ✉], Joaquín Calatayud [3], Juan A. Ballesteros-Cánovas[1,4], Adrián Escudero[5], Luis Cayuela [5], Laura Marqués[6], Marta Rueda[7], Paloma Ruiz-Benito[8], Asier Herrero [9], Cristina Aponte [10,11], Rodrigo Sagardia[12], Andrew J. Plumptre [13], Sylvain Dupire[14], Carlos I. Espinosa [15], Olga V. Tutubalina [16], Moe Myint[1], Luciano Pataro[17], Jerome López-Sáez[1], Manuel J. Macía [17,18], Meinrad Abegg [19], Miguel A. Zavala [8,20], Adolfo Quesada-Román[1,21], Mauricio Vega-Araya[22], Elena Golubeva [23], Yuliya Timokhina[23], Guillermo Bañares de Dios [3], Íñigo Granzow-de la Cerda [24] & Markus Stoffel[1,25,26]

Previous attempts to quantify tree abundance at global scale have largely neglected the role of local competition in modulating the influence of climate and soils on tree density. Here, we evaluated whether mean tree size in the world's natural forests alters the effect of global productivity on tree density. In doing so, we gathered a vast set of forest inventories including >3000 sampling plots from 23 well-conserved areas worldwide to encompass (as much as possible) the main forest biomes on Earth. We evidence that latitudinal productivity patterns of tree density become evident as large trees become dominant. Global estimates of tree abundance should, therefore, consider dependencies of latitudinal sources of variability on local biotic influences to avoid underestimating the number of trees on Earth and to properly evaluate the functional and social consequences.

[1] Climate Change Impacts and Risks in the Anthropocene (C-CIA), Institute for Environmental Sciences (ISE), University of Geneva, 66 Boulevard Carl Vogt, CH-1205 Geneva, Switzerland. [2] EiFAB-iuFOR, Universidad de Valladolid, Campus Duques de Soria s/n, 42004 Soria, Spain. [3] Departamento de Biología y Geología, Física y Química Inorgánica. ESCET, Universidad Rey Juan Carlos, C/Tulipán s/n, Móstoles, C.P. 28933 Madrid, Spain. [4] Spanish Scientific Research Council, Museo Nacional de Ciencias Naturales, Madrid, Spain. [5] Grupo de Investigación de Alto Rendimiento en Ecología de Comunidades, Departamento de Biología y Geología, Física y Química Inorgánica, ESCET, Universidad Rey Juan Carlos, C/Tulipán s/n, Móstoles, C.P. 28933 Madrid, Spain. [6] Department of Environmental Systems Science, Swiss Federal Institute of Technology (ETH Zürich), Universitätstrasse 2, 8092 Zürich, Switzerland. [7] Departamento de Biología Vegetal y Ecología, Universidad de Sevilla, C/Profesor García González s/n, 41012 Sevilla, Spain. [8] Forest Ecology and Restoration Research Group, Departamento de Ciencias de la Vida, Universidad de Alcalá, ctra., Madrid-Barcelona, km 33.4, 28805 Alcalá de Henares, Spain. [9] FisioClima CO2 Research Group, Department of Plant Biology and Ecology, Faculty of Pharmacy, University of the Basque Country, 01006 Vitoria-Gasteiz, Basque Country, Spain. [10] School of Ecosystem and Forest Sciences, The University of Melbourne, 500 Yarra Boulevard, Richmond, Victoria 3121, Australia. [11] Department of Environment and Agronomy, Centro Nacional Instituto de Investigación y Tecnología Agraria y Alimentaria, INIA-CSIC, Ctra. Coruña Km 7.5, 28040 Madrid, Spain. [12] Instituto Forestal de Chile, Sucre 2397, Ñuñoa, Santiago de Chile, Chile. [13] Key Biodiversity Area Secretariat, c/o BirdLife International, Cambridge, UK. [14] Université Grenoble Alpes, Inrae, LESSEM, 38000 Grenoble, France. [15] EcoSs_Lab, Departamento de Ciencias Biológicas, Universidad Técnica Particular de Loja, San Cayetano Alto, 110107 Loja, Ecuador. [16] Scott Polar Research Institute, University of Cambridge, Lensfield Road, Cambridge CB2 1ER, UK. [17] Departamento de Biología (Botánica), Facultad de Ciencias, Universidad Autónoma de Madrid, calle Darwin 2, Madrid, Spain. [18] Centro de Investigación en Biodiversidad y Cambio Global (CIBC-UAM), Universidad Autónoma de Madrid, Calle Darwin 2, ES–28049 Madrid, Spain. [19] Swiss Federal Institute for Forest, Snow and Landscape Research, WSL, Zürcherstrasse 111, 8903 Birmensdorf, Switzerland. [20] Instituto Franklin, Universidad de Alcalá, Calle Trinidad 1, Alcalá de Henares, 28801 Madrid, Spain. [21] Escuela de Geografía, Facultad de Ciencias Sociales, Universidad de Costa Rica, Ciudad de la Investigación, Montes de Oca, 2060 San José, Costa Rica. [22] Instituto de Investigación y Servicios Forestales (INISEFOR), Universidad Nacional de Costa Rica, 86-3000 Heredia, Costa Rica. [23] Faculty of Geography, Lomonosov Moscow State University, Moscow, Russia. [24] Real Jardín Botánico – CSIC, Plaza Murillo 2, Madrid ES-28014, Spain. [25] Department of Earth Sciences, University of Geneva, 13 rue des Maraîchers, CH-1205 Geneva, Switzerland. [26] Department F.-A. Forel for Environmental and Aquatic Sciences, University of Geneva, 66 Boulevard Carl Vogt, CH-1205 Geneva, Switzerland. ✉email: jaime.madrigal@uva.es

The most recent assessment of global tree abundance suggests that more than one trillion trees inhabit planet Earth[1]. These trees represent a massive stock of organic carbon and offer critical support for millions of species including animals, plants, fungi, lichens, and bacteria. Expressed as tree density (i.e., the number of trees per unit area), abundance represents a major structural component of natural forests linked to ecosystem functioning and energetics[1–3]. Moreover, tree abundance is a major component of diversity and can have a direct contribution to population viability and species richness in natural forests under limiting climatic conditions[4]. Thus, improving our understanding of the drivers of tree density is imperative for Earth science, ecology and conservation of forest ecosystems.

Climate is among the most conspicuous determinants of tree density. In fact, correlational evidence supports a positive, combined influence of rising temperatures (i.e., more energy) and increased water availability on the number of assembled trees per unit area[5]. Whereas evidence of this influence is hitherto inconclusive in certain biogeographical regions[6], soil fertility has been reported to be positively correlated with maximum stand density in other regions[7]. At fine scales, tree density is negatively correlated with tree size, following some density-size rules including the Yoda's law[8]. Specifically, this density-size rule implicitly depicts the critical role of competition in driving tree dynamics at the forest stand level through self-thinning constraints based on saturation of light demands by tree canopies over the course of secondary succession. It is well known that self-thinning dynamics in forests leads to a maximum carrying capacity for a given site[9]. Consequently, any biomass loss resulting from mortality of individual trees is eventually compensated by secondary growth in the remaining trees[10]. Importantly, the determinants of tree density have commonly been evaluated as independent drivers across contrasting spatial and temporal scales, such that interactive biotic influences have barely been addressed in the literature. Recent findings based on time-series of tree mortality data in a tropical forest highlighted the tight interplay between interannual climate variability and competitive interactions driving tree density[11]. Yet, it remains poorly understood whether such controls of tree density can also be recognized in the well-known, large-scale productivity patterns of tree density through competition and self-thinning dynamics operating over a set of different pools of species and biogeographical contexts.

Here, we unveil the interdependency of the main determinants of tree density at local and global extents using forest inventory data covering a large network of well-conserved forests worldwide [Fig. 1]. We hypothesize that, beyond the direct contribution of primary productivity related to well-known local and latitudinal determinants on tree density, fine-scale competitive dynamics would significantly modulate tree density patterns. We anticipate that self-thinning would be less intense in the most productive tree assemblages (i.e., tropical forests) where competition, as a major driver of natural selection in evolutionary time, promotes niche segregation[12] which in turn results in a more efficient filling of ecological space above and below the ground[13]. On the contrary, we posit that self-thinning dynamics would reduce tree density more drastically due to niche overlap among tree individuals in the most stressed, less productive environments, where environmental constraints strongly sort species with similar or convergent functional adaptations[14]. An abiotic filtering in such limiting conditions would reduce the number of co-existing species, which would in turn trigger within and between species competitiveness due to a less efficient use of nutrients at the stand level[15, 16]. If true, a latitudinal pattern of tree density would be more evident for mature forests composed of large trees because maximum forest stand density is contingent on climate conditions[17].

## Results and discussion

**Determinants of tree density at global to local scales.** We fitted a general additive mixed model (GAMM) to test for dependencies of the latitude-tree density relationship on plot-level mean diameter at breast height (DBH). Model selection using the Bayesian Information Criterion (BIC) supported the interaction between latitude and mean DBH ($BIC_{interaction} = 44{,}542.87$; $BIC_{no\_interaction} = 44{,}575.27$; $BIC_{null} = 44{,}748.20$), indicating that, across the 23 studied regions, the highest tree densities are reached in forest plots dominated by small trees irrespective of latitude [Fig. 2a], and that the latitudinal gradient of tree density is more evident towards higher mean DBH values. Interestingly, plot size was not supported as a plausible determinant of tree density ($BIC_{with\ plot\ size} = 44{,}558$; $BIC_{without\ plot\ size} = 44{,}542.87$). Different combinations of irradiance through climatic net primary productivity (NPP) with certain values of soil cation exchange capacity (CEC) a surrogate of soil fertility, induce peaks of maximum density at different latitudes. Results of a GAMM in which NPP and CEC were evaluated as a function of latitude

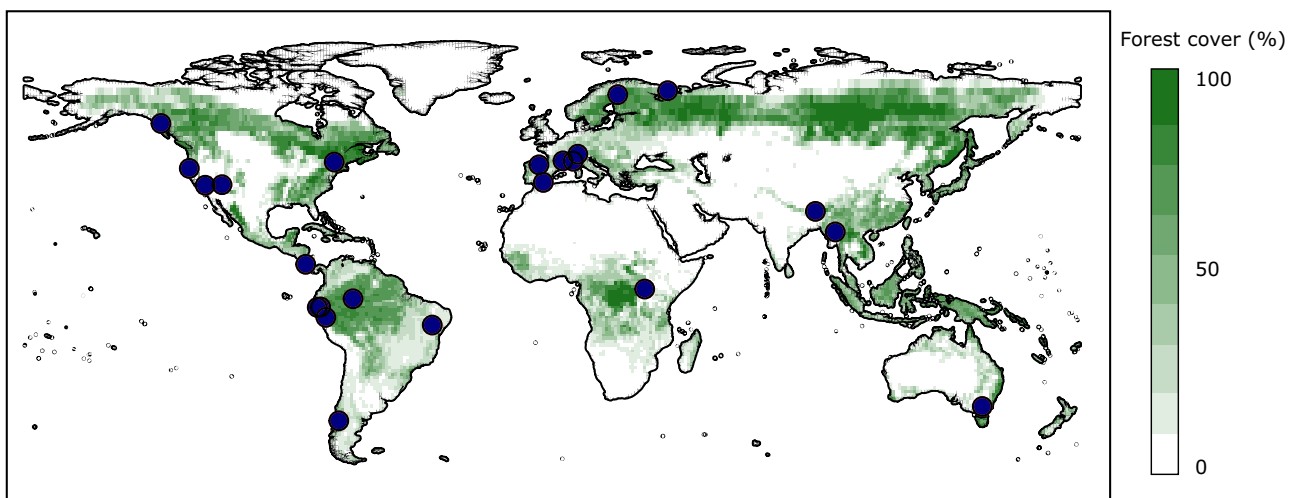

**Fig. 1 Geographic distribution of the 23 studied forests.** This map (image and every element) was fully created by the authors using averaged coordinates of the studied forest regions and forest cover data (public domain) retrieved in the FAO's website[34].

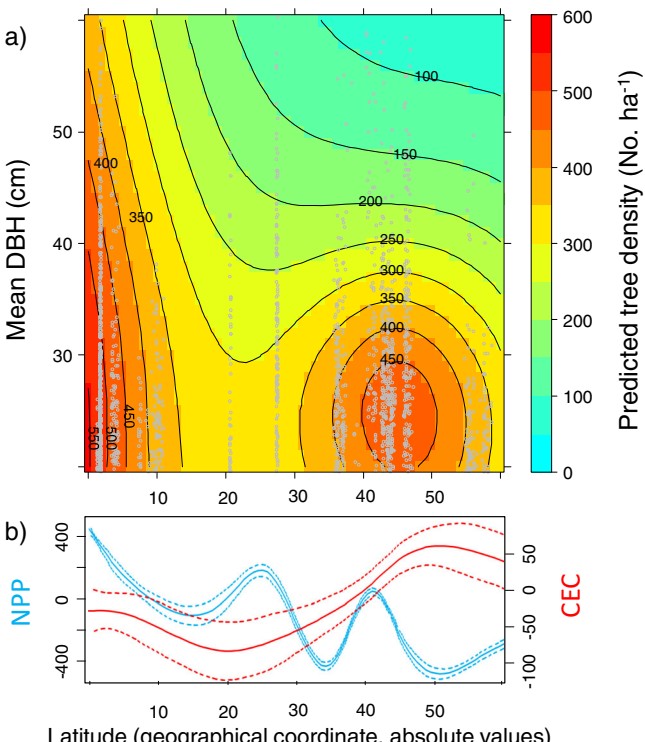

**Fig. 2 Latitude and mean DBH as predictors of tree density. a** Colored contour biplot for predictions of a generalized additive mixed model (GAMM) in which tree density (see the colored scale on the right) is expressed as a function of mean DBH (diameter at breast height) and latitude. Gray points are the data used in the GAMM. **b** predicted smoothing values of climatic Net Primary Productivity (NPP, blue lines) and Cation Exchange Capacity (CEC, red lines) as function of latitude. Solid lines represent the fitted mean values whereas dotted lines represent the 95% confidence intervals around the mean.

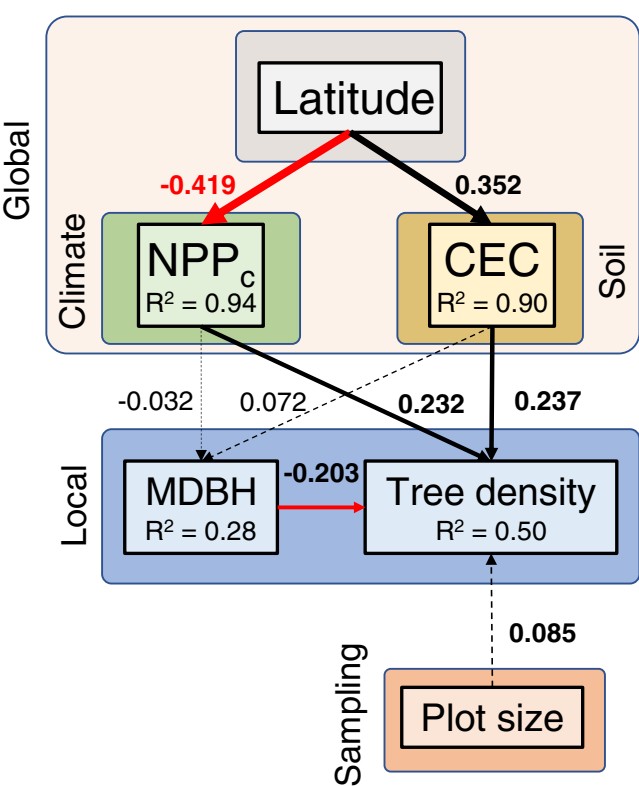

**Fig. 3 Global and local determinants of tree density.** Results of the structural equation model (SEM) fitted to tree density values taking into consideration mean diameter at breast height (MDBH), Net Primary Productivity (NPPc), Cation Exchange Capacity (CEC), and plot size, an exogenous variable included as a statistical controlling factor. Both NPP and CEC depends on latitude and are considered global factors. Solid arrows represent significant effects while dotted arrows denote non-significant relationships. Red arrows represent negative effects whereas black arrows do so for positive effects. All estimates are included in the figure as standardized estimates.

separately support the idea that, while NPP can be critical driving tree density in tropical areas, CEC can have a prevalent role driving tree density in temperate and boreal forest biomes [Fig. 2b]. These findings are in line with results of previous studies reporting climatic controls on tree abundance at global[1] and regional[18] scales. In general terms, the combination of moderate temperature and high precipitation seems to maximize the potential number of trees per unit area[1]. Nonetheless, the interplay between soil fertility and climatic NPP seems to be critical to unveil this pattern. Results of a structural equation model (SEM), defined to evaluate determinants of tree density from latitudinal sources of variability to fine grain biotic factors, evidence opposing patterns of climatic NPP and CEC. Whereas NPP is negatively associated with latitude, the opposite is true for CEC [Fig. 3]. Interestingly, both climatic NPP and CEC had positive and rather similar influences on tree density. Thus, moderate to high limiting climatic conditions in cold temperate and boreal regions could compensate for soil fertility, likely promoting elevated tree densities at high latitudes as well.

**Latitudinal patterns of tree density are contingent on plot-level mean tree size**. We also found that mean tree size affected tree density negatively, as stipulated by the Yoda's law, a pattern that has long been supported in forest science[19]. Importantly, the Bayesian information criterion supported the interactive effect between local (Mean DBH) and global (primary productivity) determinants in a linear mixed-effects model (LMM) ($BIC_{interaction} = 6980$; $BIC_{no\_interaction} = 6987$) [Fig. 4], yielding a model that explained a considerable proportion of forest density variability

(conditional $R^2 = 0.55$). This interaction suggests that plots can only support a high density of large trees when productivity is high and there is a decline in density as productivity decreases. In the high-productivity plots, the number of trees per hectare are rather similar irrespective of climate/soil conditions ($\approx 500$–$800$ trees/ha on average). Hence, if forests mature and large trees become dominant, the self-thinning dynamics seemingly operate but with different intensity depending on where trees grow along the global productivity gradient. Specifically, self-thinning dynamics seem to be more intense under severely limiting conditions than in productive environments. Consequently, the latitudinal gradient of tree density was particularly conspicuous when considering plots dominated by large trees. This means that tree density was significantly higher towards highly productive forests at low latitudes, namely tropical and equatorial rainforests. At the other end of the latitudinal gradient, in the boreal biome, tree density yielded minimum values in plots dominated by large trees. In addition, tree density was also relatively low in seasonal dry tropical and subtropical forests, where water scarcity represents a major limiting factor to tree establishment and growth. Thus, self-thinning dynamics seem to be particularly strong in areas where climate and/or soil severity limit tree performance. In these regions, abiotic filtering can result in a functional clustering tied to a selection of functional traits conferring the ability to thrive with specific environmental constraints[14]. Such a high functional overlap can be expected to result in more intense and

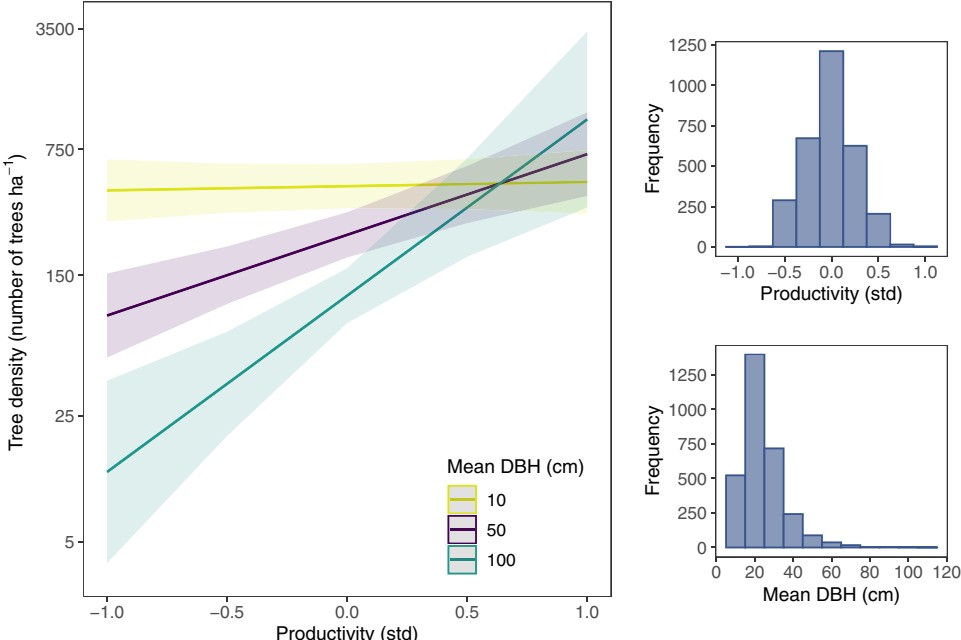

**Fig. 4 The productivity-tree density relationship is contingent upon mean DBH.** Graphical representation of tree density as a function of the interaction between mean tree size (Mean DBH) and productivity calculated as 0.232NPP + 0.237CEC (being NPP the climatic Net Primary Productivity, and CEC the Cation Exchange Capacity). Shadowed areas represent the 95% confidence intervals of model predictions. The right panels display the distributions of mean DBH and productivity, respectively. Results support the idea that the positive influence of productivity on tree density reinforces towards higher values of mean tree size.

competitive interactions with the course of forest development and fine-scale (competitiveness) dynamics[15]. Likewise, temperate forests were reported to undergo important self-thinning events during forest stand development and succession according to density-size relationships[20]. In tropical forests, self-thinning dynamics have been observed to affect tree density along fine-scale environmental gradients depending on dry season length and frequency of natural disturbances[21]. Nonetheless, tree density in tropical rainforests tends to be high even for big trees, which might be justified in part by high productivity[22] but also by the huge functional trait dispersion, likely forced in evolutionary time by competition to induce niche segregation and natural selection. High niche segregation has been reported in these type-forests in relation to light and water niche axes[23].

**Implications and limitations**. Our study underlines the crucial role played by climate and soil fertility on global gradients of tree density in natural forests but also stipulates that global tree density is significantly modulated by competitiveness. Although a simplified pool of potential drivers of abundance was used, our findings point to the important role of fine-scale determinants, such as mean tree size, in modulating global patterns of tree density associated with climatic and soil productivity. Further research should focus on disentangling more specific and complex interactions between latitudinal sources of variability and local context variables so as to elucidate mechanisms through which biotic interactions (i.e., competition) can alter global tree density patterns in more detail. For instance, dealing separately with precipitation and soil nutrients might help to unveil the actual links between water availability and nutrients with competition and forest dynamics along the latitudinal gradient. Whereas water scarcity can play a major role in dry forests, nutrient availability becomes a key driver in tropical rainforests, where abundant precipitation is responsible for the leaching loss of nutrients. Similarly, potential signals of local disturbances and historical contingencies, but also methodological choices (e.g., the inclusion of trees with a DBH ≥

10 cm only) deserve future attention. For the moment, even if the data used in this study cover most of the size spectrum, and DBH cutoffs represent standardized methods in the national forest inventories (NFI), further efforts should be made to extend monitoring protocols to trees smaller than 10 cm DBH (as is the case in some NFI). Whereas these sources of uncertainty might potentially blur signals of tree density patterns, drivers used in this study accounted for a considerable portion of variability in tree density, and thus support the importance of the interplay between productivity and competition.

Modulation of global tree density by mean tree sizes will have major implications on the way carbon is stored in forests through the formation of different size structures under specific climatic and/or soil productivity gradients. Thus, our findings also call for such dependencies to be considered more thoroughly in global assessments of tree density and the ensuing functional consequences related to forest functions and the provisioning of services to human societies.

## Materials and methods
**Forest inventory data and selection criteria**. We used information on tree density recorded in more than 3000 well-conserved forest plots distributed across protected natural areas in 23 regions worldwide. Specifically, data were obtained from North (United States), Central (Costa Rica), and South (Ecuador, Brazil, Bolivia, Peru, and Chile) America, Africa (Uganda), Oceania (Australia), Asia (eastern Russia, Bhutan, and Myanmar), and Europe (Sweden, Switzerland, France, and Spain) (see Table 1 for details). All forest plots used here were retrieved from national forest inventories or obtained from research projects. Sampling plots can be either circular or rectangular (depending on the region considered) with identical sizes within each forest region. We only considered forest regions for which a minimum of 30 forest plots were available to ensure robustness in the statistical analysis. To exclude anthropogenic impacts, we only considered natural forests within protected areas lacking evidence of recent disturbance. Further information on the database can be found in Madrigal-González et al.[4].

**Biotic and abiotic data**. Every tree exceeding 10 cm in DBH from each sampling plot was identified at the species level or classified as a morphospecies (in the case of Bhutan, Myanmar, Ecuador, and Peru). For each region, tree density was estimated as the number of standing trees per hectare and mean tree size as plot-level

**Table 1 Descriptive geographic and sampling information for each of the 23 forest regions.**

| Country | Region | Longitude (°) | Latitude (°) | Elevation (m.a.s.l.) | No. Plots | Plot Size (m²) |
|---|---|---|---|---|---|---|
| Russia | Kola | 39.19 | 67.58 | 199.045 | 28 | 400 |
| US | Sequoia National Park | −118.41 | 36.20 | 2274.69 | 132 | 168.33 |
| US | Great Canyon National Park | −112.12 | 36.26 | 2250.94 | 68 | 168.33 |
| Sweden | Northern Sweden | 22.31 | 66.17 | 141.40 | 101 | 1256 |
| Spain | Sierra Nevada National Park | −3.72 | 38.10 | 1733.92 | 56 | 1963.49 |
| Switzerland | Alps | 8.89 | 46.39 | 1496.74 | 234 | 500.34 |
| Bhutan | Toepisa | 89.78 | 27.52 | 2371.11 | 160 | 500.34 |
| US | Alaska | −133.16 | 56.30 | 874.33 | 491 | 168.33 |
| US | New York | −74.47 | 43.66 | 1815.22 | 197 | 168.33 |
| Brazil | Bahia | −40.90 | −9.99 | 884.37 | 106 | 400 |
| Ecuador | Western Ecuador | −80.19 | −4.22 | 580.62 | 48 | 400 |
| Australia | Victoria | 146.80 | −37.13 | 468.11 | 35 | 900 |
| France | Mercantour National Park | 7.12 | 44.13 | 1806 | 61 | 706 |
| Chile | Northern Patagonia | −72.63 | -41.59 | 361.22 | 109 | 500 |
| France | Cévennes National Park | 3.65 | 44.26 | 1123.82 | 98 | 706 |
| Spain | Fuentes Carrionas Natural Park | −4.54 | 42.93 | 1316 | 117 | 1963.49 |
| US | Klamath Forest | −123.87 | 41.78 | 787.43 | 74 | 168.33 |
| Ecuador | Podocarpus National Park | −79.01 | −4.12 | 1959 | 30 | 1000 |
| Peru | Río Abiseo National Park | −77.38 | −7.64 | 1948.80 | 30 | 1000 |
| Myanmar | Wetphuyay | 96.54 | 20.72 | 1095.66 | 62 | 500.34 |
| Uganda | Forest Reserve | 31.54 | 1.76 | 1042.00 | 622 | 100 |
| Bolivia | Madidi National Park | −67.94 | −14.21 | 656.25 | 44 | 1000 |
| Costa Rica | Costa Rica | −84.15 | 9.87 | 933.97 | 44 | 1000 |

Values for longitude, latitude, elevation, number of plots and plot size are shown next to names of regions and countries.

mean DBH. We interpret the negative relationship between mean DBH and tree density as a static realization of self-thinning dynamics over time in line with expectations of Yoda's law[8]. Geographical coordinates were obtained for each plot using global positioning systems (GPS). The climatological net primary productivity (NPP) index[24] was retrieved as a nonlinear combination of temperature and precipitation following the equations of the Miami model[25] (see(1) for details) where NPP increases with rising temperature and precipitation up to a saturation of 3000 g dry matter m$^{-2}$ year$^{-1}$. Soil cation exchange capacity (CEC, cmolc/kg) was used here as a surrogate of soil fertility and retrieved from the SoilGrids system, which provides global gridded soil information at 250 m resolution accounting for climate, land cover and topography variability[26]. We calculated an all-inclusive productivity variable combining NPP and CEC proportionally to their estimated effects on tree density in the structural equation model (0.232, 0.237 respectively; unstandardized coefficients; see statistical analyses below).

**Statistics and reproducibility.** To explore the potential contribution of global and local drivers to tree density, we first fitted a generalized additive mixed model (GAMM[27]) where tree density was the response variable and mean tree size and absolute latitude were included as potential covariates. GAMM is a flexible method that allows modeling complex non-linear relationships of a response variable and one or several covariates. To account for site-to-site variability, a nested random factor of plots within tracks within regions was included in the model. Tracks are spatial clustering of sampling plots within regions. For instance, in the US, sampling plots are spatially clustered in groups of four plots around each fixed point of the grid considered across the country.

We then applied a structural equation model (SEM[28]) to test the relative contribution of each environmental determinant on tree density. Following the theoretical framework defined in Fig. 3, latitude was included as an exogenous variable controlling variability in NPP and CEC, which in turn are considered to affect tree density at such a broad scale. Mean tree size and plot size were included in the SEM as exogenous local variables that could affect tree density. We applied mixed-effect models with random intercepts following the previous criterion (i.e., tracks within regions) for each causal relationship inside our SEM. For the final model selection, we used Fisher's C information criterion.

To evaluate how local mean tree size modulates the global productivity effect on tree density, we used linear mixed-effects models with the following structure:

$$Y = Xa + Zb + \varepsilon \quad (1)$$

where Y represents tree density across forest regions, a is the vector of parameters in the fixed effects term, that includes plot size and the interaction between the all-inclusive productivity and mean tree size, b is the vector of parameters of the random effects (i.e., nested structure of tracks within a forest region), X and Z are regression matrices of fixed and random effects, respectively, and $\varepsilon$ is the within-group error component. For all the above-mentioned analyses, tree density was log-transformed to meet the assumption of linear models of normally distributed errors and homoscedasticity. All continuous predictor variables were standardized to

improve the interpretability of effect sizes and interactions. We built contrasting models differing in their fixed effects using a backward selection procedure starting with a full model that includes all hypothesized variables and the interaction (productivity × mean_DBH). In all cases, model comparison was conducted with the BIC. We used BIC as it performs better than the Akaike Information Criterion when dealing with large datasets[29]. First, we eliminated the interaction between productivity and mean tree size and compare this model with the initial full model. Second, we kept the interaction between productivity and mean tree size and removed plot size. Lastly, we compared this model with a null model. All the analyses were performed with the R statistical software version 4.1.1[30]. Generalized additive mixed models were computed using the *gamm* function from the *mgcv* R package[31]. SEM model were analyzed using the *psem* function from the *piecewiseSEM* R package[32]. Linear mixed-effects model were conducted using the *lme* function from the *nlme* R package[33].

**Reporting summary.** Further information on research design is available in the Nature Portfolio Reporting Summary linked to this article.

## Data availability

The forest data used can be accessed in https://figshare.com/articles/dataset/Forest_csv/13072211. Source data for Fig. 2 can be found in Supplementary Data 1. Source data for Fig. 4 can be found in Supplementary Data 2.

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

## Acknowledgements

JMG was funded by the Research Commission of the Swiss National Park, Swiss Academy of Sciences SCNAT (project FLUCTREE- 02/2020). J.C. thanks to Spanish Ministerio de Ciencia e Innovación (Project UNIPER, PID2020-114851GA-100). J.B.C. was funded by the Talento fellowships, Comunidad de Madrid (project INOVA-RISK, 2020-T1/AMB-19913). AE thanks the European Social Fund managed by the Regional Government of Madrid (Remedinal TE-CM: S2018/EMT-4338). L.C. and M.J.M. were partly funded through projects CGL2013-45634-P, CGL2016-75414-P, and PID2019-105064GB-I00. L.M. was funded by the Swiss National Science Foundation (PCEFP2_181115) and the Margarita Salas Postdoctoral Fellowship from Universidad de Alcalá (MIU, Spain). A.H. was supported by the Basque Country Government funding support to FisioClimaCO₂ (IT1682-22) research group. GBD was funded by the Spanish Ministry of Education (MINEDU; FPU14/05303) and the Regional Government of Madrid (Remedinal TE-CM: S2018/EMT-4338).

## Author contributions

Conceptualization: J.M.G., J.C., M.S., A.E. Data acquisition: J.M.G., J.C., J.B.C., A.E., L.C., L.M., M.R., P.R.B., A.H., C.A., R.S., A.P., S.D., C.E., O.T., M.M., L.P., J.L.S., M.J.M., M.A., M.A.Z., A.Q.R., M.V.A., E.G., Y.T., G.B.D., I.G.C., M.S. Methodology: J.M.G., J.C., A.E., L.M. Investigation: J.M.G., J.C., M.S. Visualization: J.M.G., M.S. Funding acquisition: J.M.G., J.C., M.S., A.E. Project administration: J.M.G., J.C., M.S., A.E. Supervision: J.M.G., M.S. Writing – original draft: J.M.G., J.C., J.B.C., A.E., L.M. Writing – review & editing: J.M.G., J.C., J.B.C., A.E., L.C., L.M., M.R., P.R.B., A.H., C.A., R.S., A.P., S.D., C.E., O.T., M.M., L.P., J.L.S., M.J.M., M.A., M.A.Z., A.Q.R., M.V.A., E.G., Y.T., G.B.D., I.G.C., MS.

## Competing interests

The authors declare no competing interests.
