## [Peer Review File · Communications Biology]

Reviewers' comments:

Reviewer #1 (Remarks to the Author):

This is an interesting and succinct paper describing global trends in tree density. I think that the results are important to publish, but that the uncertainties, particularly with respect to bias and covariation of confounding factors is missing.

I would like to see Table S1 include a map of the plot locations.

I would like to see some discussion on uncertainties. The 10-cm cut off for minimum tree diameter excludes a significant component of demographic processes. How does this affect the analysis and trends with latitude?

Also, site history and disturbance interactions with demographic processes is not discussed and would contribute to uncertainties. These processes would also affect the latitudinal gradients and correlations with tree density.

Reviewer #2 (Remarks to the Author):

The Madrigal-González et al manuscript "Global patterns of tree density are contingent upon local determinants in world's natural forests" presents an analysis of environmental drivers of tree density in >3,000 plots sampled across 23 global study sites. A combination of mixed effects modelling and structural equation models are used to illustrate how climate (net primary productivity; NPP) and soil (cation exchange capacity; CEC) influence tree density, but in contrasting ways depending on latitude. An interactive effect of (global) productivity and (local) tree size is also found though, with a high density of small trees irrespective of NPP, but a high density of large trees only supported when NPP is high and density declines as NPP decreases.

These results are important because it is one of the few studies to show interactive effects of biotic drivers on tree density. They also highlight how local context can alter the effect of global drivers of tree density such as NPP. The dataset is impressive and the analyses are largely robust, although I have some criticism related to the small subset of potential variables that are actually analysed (see below). The story also needs some clarification in relation to the interpretation of both the global and local determinants of tree density. I also have some suggestions on how to improve the format and flow of the main text.

Specific comments:

Ln63. Change to "in the world's". This comment also applies to the title.

Ln68. Change to "avoid underestimating the number"

Ln69. Delete "ensued"

Ln72. The first three paragraphs should be in a section called "Introduction". The remaining sections can form a section called "Results and Discussion". The Methods should be moved from the supplementary material to the end of the main text.

Ln76. Replace the hash sign with the word "number"

Ln78. It is unclear how abundance can have "a direct contribution" to species richness. Abundance only affects the evenness component of diversity, not the richness, which is purely determined by presence-absence. I suspect you mean that dominant trees squeeze out the space available for additional species, but this should be clarified.

Ln88. Clarify exactly what Yoda's law means.

Ln95. Delete "eventually"

Ln96. Change "recognized to" to "recognized in"

Ln101. The article is short and there is space for more display items. Move Table S1 to the main text and cite it here as Table 1. Note some incorrect spellings (e.g. Sequoia, Sweden) and provide the name of the Ugandan national park. The suggestion of reviewer #1 to include a map of the study sites is optional and would need to be a separate figure because it cannot be combined with the table.

Ln103. Is it just latitude that controls NPP and CEC? Surely longitude or topography should have a role to play, e.g. I imagine coastal or mountainous regions would have a very different NPP/CEC compared to what you might otherwise predict for a low-lying patch of land at the same latitude.

Ln106. Change "opposite" to "contrary"

Ln115. Change to "of the latitude"

Ln123. This is a slight oversimplification of the trends in Figure 1b. There is a peak of NPP at 25° latitude, which is higher than the one at 40° and does not coincide with high tree density. Peak CEC also does not coincide with any peak of NPP and neither coincide directly with the peak tree density in the temperate region. No need to water this down too much, but be a bit more open about the imperfect fit here.

Ln131. This finding supports the poor fit of the patterns in Figure 1b, where you can see an overall decline in NPP (despite the peaks) and an overall increase in CEC (despite the slight upcurve in tropical areas). Ultimately, you may be better off noting for Figure 1b that NPP could be the key driver of peak tree density in tropical areas, while CEC may be more relevant in temperate zones (stating the relevant latitudes for each one to be clear on what you mean by each as well).

Ln134. This is a really nice point and a great way of explaining the apparent contradiction in the preceding sentence. Can you add a similar point for tropical regions?

Ln141. I find this statement to mischaracterise the result somewhat. It is not so much that the positive influence "fades out" when plots are dominated by small trees...density is consistently high for all levels of productivity in that case. In contrast, plots can only support a high density of large trees when productivity is high and there is a decline in density as productivity decreases.

Ln163. Change to "tends"

Ln170. It would be good to develop the link between tree size and competitiveness a little earlier on.

Ln172. Change to "associated with"

Ln177. Change to "management in the face of global warming"

Methods:

Ln65. Provide a summary of how plot size varied across all sites, e.g. I assume the column in Table S1 is just the mean, so include SD and range as well. Did you correct for this in some way in your analyses or test whether it could influence the results, given that bigger plots could have more species and are more likely to include bigger trees? For example, plot size could be included in models as a covariate.

Ln66. Provide some logic for the choice of these study sites and not others.

Ln90. Do NPP and CEC really summarise all the possible climatic and soil variables that could be influencing tree density? What about temperature, precipitation, aridity, etc. as other climate variables (freely available through online databases such as WorldCLIM). I guess soil moisture, pH, permeability, etc. will also be important soil variables, though perhaps more challenging to acquire en masse. If no changes are made to the modelling framework, some discussion of the limitations of this approach would be needed and the possibility for latent variables to influencing your outcomes.

Ln99. What are "tracks"?

Ln103. Figure S2 is effectively repeated in Figure 2 and so does not need to be included as a separate figure. Just note that all the arrows in the figure are the a priori hypothesised or theoretical pathways. Given the changes suggested above, this should make supplementary material redundant for the article.

Ln104. Ok, it's good that plot size is included in the SEM, but that accentuates the need to consider it in the GAMM approach as well.

Ln106. Why just tracks within regions? What happened to the plots within tracks within regions structure of the GAMM? Try to use a more consistent analytical framework across the two modelling approaches.

Ln111. Can you clarify what you mean by the "vector of fixed effects" here? Is this some aggregated term and if so, how did you calculate it? Or did you actually have separate main and interactive effects in the model and if so, can you spell the full model out more clearly?

Ln112. Unclear what you mean by "the all-inclusive" here.

Ln118. It's unclear why you need to use BIC...what is the full set of models that you are choosing between? I think this confusion arises from the "all inclusive" way you describe the fixed effects of your LMEs...more clarity is needed here.

RESPONSE LETTER

Reviewers' comments:

Reviewer #1 (Remarks to the Author):

This is an interesting and succinct paper describing global trends in tree density. I think that the results are important to publish, but that the uncertainties, particularly with respect to bias and covariation of confounding factors is missing.

I would like to see Table S1 include a map of the plot locations.

Following the reviewer's advice, we have included a map showing the location of the study sites (Fig. 1) and moved Table S1 to the main text (now Table 1).

I would like to see some discussion on uncertainties. The 10-cm cut off for minimum tree diameter excludes a significant component of demographic processes. How does this affect the analysis and trends with latitude?

Please note that DBH cut offs are standard procedures in forest inventories. Also, tree size data cover most of the size spectrum. However, we acknowledge that minimum DBH can affect tree density assessments especially with tree species with resprouting ability, where a given individual can present several low-size stems. We included a few lines: “*Similarly, potential signals of local disturbances and historical contingencies, but also methodological choices (e.g., the inclusion of trees with a DBH ≥ 10 cm only) deserve future attention. For the moment, even if the data used in this study cover most of the size spectrum, and dbh cutoffs represent standardized methods in the national forest inventories, further efforts should be made to extending monitoring protocols to trees smaller than 10 cm DBH (as it is otherwise the case in some NFI).*” (L192-197).

Also, site history and disturbance interactions with demographic processes is not discussed and would contribute to uncertainties. These processes would also affect the latitudinal gradients and correlations with tree density.

We agree with the reviewer that site history and disturbance are important local factors affecting demographic processes. While these sources of uncertainty have the potential to partially swamp any signal of tree density patterns, the drivers used in this study accounted for a considerable portion of the variability in tree density, thus supporting the importance of the interplay between productivity and competition. Otherwise, such information is not available in general. Note that the majority of the sites of the study are placed in remote areas where human influences are almost absent for at least the last 80 years. Data included in the National Forest Inventory databases are still at early stages that result insufficient to cover the site history because most of them only cover a static picture with a single field survey. For this reason, it is not possible for us to incorporate this information in the models. We have included a sentence in the last part of the discussion to evidence this lack of information and the necessity to account for

long-term research monitoring schemes in forest ecosystems worldwide (L192-197).

Reviewer #2 (Remarks to the Author):

The Madrigal-González et al manuscript “Global patterns of tree density are contingent upon local determinants in world’s natural forests” presents an analysis of environmental drivers of tree density in >3,000 plots sampled across 23 global study sites. A combination of mixed effects modelling and structural equation models are used to illustrate how climate (net primary productivity; NPP) and soil (cation exchange capacity; CEC) influence tree density, but in contrasting ways depending on latitude. An interactive effect of (global) productivity and (local) tree size is also found though, with a high density of small trees irrespective of NPP, but a high density of large trees only supported when NPP is high and density declines as NPP decreases.

These results are important because it is one of the few studies to show interactive effects of biotic drivers on tree density. They also highlight how local context can alter the effect of global drivers of tree density such as NPP. The dataset is impressive and the analyses are largely robust, although I have some criticism related to the small subset of potential variables that are actually analysed (see below). The story also needs some clarification in relation to the interpretation of both the global and local determinants of tree density. I also have some suggestions on how to improve the format and flow of the main text.

Thank you very much for the positive comments on our manuscript. Please, below you can find responses to every comment/question addressed.

Specific comments:

Ln63. Change to “in the world’s”. This comment also applies to the title.

Done (L63 and title)

Ln68. Change to “avoid underestimating the number”

Done (L69)

Ln69. Delete “ensued”

Done.

Ln72. The first three paragraphs should be in a section called “Introduction”. The remaining sections can form a section called “Results and Discussion”. The Methods should be moved from the supplementary material to the end of the main text.

Done.

Ln76. Replace the hash sign with the word “number”

Done (L78)

Ln78. It is unclear how abundance can have “a direct contribution” to species richness. Abundance only affects the evenness component of diversity, not the richness, which is purely determined by presence-absence. I suspect you mean that dominant trees squeeze out the space available for additional species, but this should be clarified.

We agree that the intention of the sentence was not clear enough. Abundance is critical to species richness because abundance is critical to population viability. We have rephrased the sentence accordingly, which now reads: “Moreover, tree abundance is a major component of diversity and have a direct contribution to population viability and species richness in natural forests under limiting climatic conditions” (L80-81).

Ln88. Clarify exactly what Yoda’s law means.

We have added a sentence to clarify the Yoda’s law meaning: “Specifically, this density-size rule implicitly depicts the critical role of competition driving tree dynamics at the forest stand level through self-thinning constraints based on saturation of light demands by tree canopies over the course of secondary succession.” (L92-92)

Ln95. Delete “eventually”

Done.

Ln96. Change “recognized to” to “recognized in”

Done (L101)

Ln101. The article is short and there is space for more display items. Move Table S1 to the main text and cite it here as Table 1. Note some incorrect spellings (e.g. Sequoia, Sweden) and provide the name of the Ugandan national park. The suggestion of reviewer #1 to include a map of the study sites is optional and would need to be a separate figure because it cannot be combined with the table.

Done. We have moved table S1 to the main text (it is now Table 1)

Ln103. Is it just latitude that controls NPP and CEC? Surely longitude or topography should have a role to play, e.g. I imagine coastal or mountainous regions would have a very different NPP/CEC compared to what you might otherwise predict for a low-lying patch of land at the same latitude.

The reviewer is right in that NPP and CEC might show patterns associated with longitude and topography. Yet, it is out of the scope of this paper to search for the best predictors for these two potential drivers influencing tree density. In addition, note that random factors might indirectly account for the effects of longitude or even topography on NPP and CEC, so in order to simplify the analyses and the presentation of results, we strongly believe it is better not to include these effects in the SEM.

Ln106. Change “opposite” to “contrary”

Done (L112).

Ln115. Change to “of the latitude”

Done (L123)

Ln123. This is a slight oversimplification of the trends in Figure 1b. There is a peak of NPP at 25° latitude, which is higher than the one at 40° and does not coincide with high tree density. Peak CEC also does not coincide with any peak of NPP and neither coincide directly with the peak tree density in the temperate region. No need to water this down too much, but be a bit more open about the imperfect fit here.

Ln131. This finding supports the poor fit of the patterns in Figure 1b, where you can see an overall decline in NPP (despite the peaks) and an overall increase in CEC (despite the slight upcurve in tropical areas). Ultimately, you may be better off noting for Figure 1b that NPP could be the key driver of peak tree density in tropical areas, while CEC may be more relevant in temperate zones (stating the relevant latitudes for each one to be clear on what you mean by each as well).

Thank you for bringing this up. We have rephrased the paragraph as to introduce the potential influences of other latitudinal pressures on tree density patterns. The new paragraph now reads: “Different combinations of irradiance through climatic net primary productivity (NPP) with certain values of soil cation exchange capacity (CEC) a surrogate of soil fertility, induce peaks of maximum density at different latitudes. Results of a GAMM in which NPP and CEC were evaluated as function of latitude separately support the idea that, while NPP can be critical driving tree density in tropical areas, CEC can have a prevalent role driving tree density in temperate and boreal forest biomes [Fig. 2b]” (L131-136).

Ln134. This is a really nice point and a great way of explaining the apparent contradiction in the preceding sentence. Can you add a similar point for tropical regions?

Thank you for the comment. We believe that the new paragraph shown in the previous comment includes what it is requested in this one.

Ln141. I find this statement to mischaracterise the result somewhat. It is not so much that the positive influence “fades out” when plots are dominated by small trees...density is consistently high for all levels of productivity in that case. In contrast, plots can only support a high density of large trees when productivity is high and there is a decline in density as productivity decreases.

Thank you very much for this comment. We have rephrased the sentence as follows: “This interaction suggests that plots can only support a high density of large trees when productivity is high and there is a decline in density as productivity decreases” (see L152-154).

Ln163. Change to “tends”

Done (L175).

Ln170. It would be good to develop the link between tree size and competitiveness a little earlier on.

We have included a sentence discussing this idea in the second paragraph of the introduction (L90-92).

Ln172. Change to “associated with”

Done (L184)

Ln177. Change to “management in the face of global warming”

We rephrased this part, so this sentence has completely changed.

Methods:

Ln65. Provide a summary of how plot size varied across all sites, e.g. I assume the column in Table S1 is just the mean, so include SD and range as well. Did you correct for this in some way in your analyses or test whether it could influence the results, given that bigger plots could have more species and are more likely to include bigger trees? For example, plot size could be included in models as a covariate.

Plots within a region are all the same size as established in the National Forest Inventories protocols. Similarly, plots obtained in research projects are strictly the same size. In Lines 214-217 of the current version (Materials and Methods section) we mention this explicitly and emphasize that plot size varies only among regions, which is the reason to include plot size in the statistical analyses: “All forest plots used here were retrieved from national forest inventories or obtained from research projects. Sampling plots can be either circular or rectangular (depending on the region considered) with identical sizes within each forest region”

Ln66. Provide some logic for the choice of these study sites and not others.

The logic for the inclusion of these plots is to have the widest possible representation of principal forest biomes on Earth considering, at the same time, the enormous limitations imposed by data availability (not only existing data but also access depending on legal and personal limitations). We rephrase the sentence to read as: “ [...] to cover as much as possible the main forest biomes on Earth” (L64-66).

Ln90. Do NPP and CEC really summarise all the possible climatic and soil variables that could be influencing tree density? What about temperature, precipitation, aridity, etc. as other climate variables (freely available through online databases such as WorldCLIM). I guess soil moisture, pH, permeability, etc. will also be important soil variables, though perhaps more challenging to acquire en masse. If no changes are made to the modelling framework, some discussion of the limitations of this approach would be needed and the possibility for latent variables to influencing your outcomes.

We agree that a myriad of climatic-edaphic factors can have a role in latitudinal tree density patterns. On the one hand, NPP is a combination of precipitation and temperature following the Miami model equations. On the other, CEC is mostly related to soil fertility and is closely related to pH and available nutrients. Of course, these two variables are very synthetic ways to define climatic/edaphic variability latitudinally speaking, but we just want to approach the big picture of productivity gradients in our analyses to avoid an exhaustive searching which is, otherwise, out of our goals. Following your advice, we include a short paragraph at the end of the discussion to highlight the potential limitations of simplifying the number of potential drivers of tree density latitudinally: “Although a simplified pool of potential drivers of abundance was used, our finding points to the important role of fine-scale determinants such as mean tree size in modulating global patterns of tree density associated with climatic and soil productivity. Further research should be focused on untangling more specific and complex

interactions between latitudinal sources of variability and local context variables so as to elucidate more in-depth the mechanisms through which biotic interactions, i.e., competition, can alter global tree density patterns. For instance, dealing separately with precipitation and soil nutrients might help us to unveil the actual links between water availability and nutrients with competition and forest dynamics along the latitudinal gradient. While water scarcity can play a major role in dry forests, nutrient availability might ostent a prevalence in tropical rainforests where abundant precipitation is responsible for leaching loss of nutrients.” (L182-192).

Ln99. What are “tracks”?

Tracks are spatial clustering of sampling plots within regions. For instance, in the US, sampling plots are spatially clustered in groups of four plots around each fixed point of the grid considered across the country. This clustering must be considered in the models to be nested within regions. That is why we include *Track* within *Region* as the random factor in the linear mixed-effects models and the GAMM. We added this clarification also in the main text (L247-249)

Ln103. Figure S2 is effectively repeated in Figure 2 and so does not need to be included as a separate figure. Just note that all the arrows in the figure are the a priori hypothesised or theoretical pathways. Given the changes suggested above, this should make supplementary material redundant for the article.

We have removed Figure S2 and therefore the supplementary material from the manuscript.

Ln104. Ok, it's good that plot size is included in the SEM, but that accentuates the need to consider it in the GAMM approach as well.

Yes, absolutely. We included plot size in the GAMM and the BIC did not support its inclusion in the final model (BIC with Plot size=44558 vs BIC without Plot size=44542). We have included this result in lines 129-130.

Ln106. Why just tracks within regions? What happened to the plots within tracks within regions structure of the GAMM? Try to use a more consistent analytical framework across the two modelling approaches.

Plots are the spatial units in this framework, so there is not any replicate within plots that justifies clustering. In other words, there is only one measurement per plot and so plot can't be a random term because it does not imply pseudoreplication.

Ln111. Can you clarify what you mean by the “vector of fixed effects” here? Is this some aggregated term and if so, how did you calculate it? Or did you actually have separate main and interactive effects in the model and if so, can you spell the full model out more clearly?

Thank you very much for this comment. We have noticed it was unclear what we meant with *vector*. We have rephrased the sentence and clarified that vector represents the set of parameters linked to the fixed-effects and the random-effects terms, respectively. The sentence now reads: “...where *Y* represents tree density across forest regions, *a* is the vector of parameters in the fixed effects term, that includes plot size and the interaction between the all-inclusive productivity and

mean tree size, b is the vector of parameters of the random effects (i.e., nested structure of tracks within a forest region), X and Z are regression matrices of fixed and random effects, respectively, and ε is the within-group error component.” (L260-264)

Ln112. Unclear what you mean by “the all-inclusive” here.

We have clarified what productivity is in lines 236-239:” We calculated an all-inclusive productivity variable combining NPP and CEC proportionally to their estimated effects on tree density in the structural equation model (0.232, 0.237 respectively; unstandardized coefficients; see statistical analyses below)”.

Ln118. It’s unclear why you need to use BIC...what is the full set of models that you are choosing between? I think this confusion arises from the “all inclusive” way you describe the fixed effects of your LMEs...more clarity is needed here.

We have included a paragraph in lines 266-271 to explain in detail how and why we use the BIC in the model selection procedure: “All continuous predictor variables were standardized to improve the interpretability of effect sizes and interactions. We used the Bayesian Information Criterion (BIC) for the evaluation of the fixed effects using a backward selection procedure starting with a model that includes all the hypothesized variables and the interaction. First, we eliminated the interaction between productivity and mean tree size and compare this model with the initial model which includes the interaction. Second, we kept the interaction between productivity and mean tree size and removed Plot size. This model was compared with the initial model using the BIC to test the validity of Plot size as a predictor of tree density. We used BIC as it performs better than the Akaike Information Criterion (AIC) when dealing with large data sets(29).”.

REVIEWERS' COMMENTS:

Reviewer #2 (Remarks to the Author):

Madrigal-González et al. have satisfactorily addressed all the comments of myself and the other reviewer in their revision. I appreciated the new text acknowledging the limited pool of climatic and environmental variables included here, which could be addressed in future research. I also found the new explanations of model structure, selection protocols, and model interpretation a big improvement. The new map figure is also a welcome addition. I have only a couple of very minor typographical recommendations remaining.

Ln134. Change to "as a function of"

Ln195. Change to "and DBH cutoffs represent standardized methods in the national forest inventories (NFI), further efforts should be made to extend monitoring protocols to trees smaller than 10 cm DBH (as is the case in some NFI)"